# The Cumulative Effect of Expanding the Breadth and Scope of Coverage for Substance Use Disorder Treatment on Behavioral Health Acute Inpatient Admissions: Evidence from Virginia Medicaid

**DOI:** 10.3390/ijerph21060777

**Published:** 2024-06-14

**Authors:** Shiva Salehian, Peter Cunningham, Andrew Barnes, Shoou-Yih Daniel Lee

**Affiliations:** Department of Health Policy, School of Population Health, Virginia Commonwealth University, Richmond, VA 23219, USA; pjcunningham@vcu.edu (P.C.); abarnes3@vcu.edu (A.B.); leesd@vcu.edu (S.-Y.D.L.)

**Keywords:** Medicaid, mental health, acute inpatient care

## Abstract

We evaluated the impact of Medicaid policies in Virginia (VA), namely the Addiction and Recovery Treatment Services (ARTS) program and Medicaid expansion, on the number of behavioral health acute inpatient admissions from 2016 to 2019. We used Poisson fixed-effect event study regression and compared average proportional differences in admissions over three time periods: (1) prior to ARTS; (2) following ARTS but before Medicaid expansion; (3) post-Medicaid expansion. The number of behavioral health acute inpatient admissions decreased by 2.6% (95% CI [−5.1, −0.2]) in the first quarter of 2018 and this decrease gradually intensified by 4.9% (95% CI [−7.5, −2.4]) in the fourth quarter of 2018 compared to the second quarter of 2017 (beginning of ARTS) in VA relative to North Carolina (NC). Following the first quarter of 2019 (beginning of Medicaid expansion), decreases in VA admissions became larger relative to NC. The average proportional difference in admissions estimated a decrease of 2.7% (95% CI, [−4.1, −0.8]) after ARTS but before Medicaid expansion and a decrease of 2.9% (95% CI, [−6.1, 0.4]) post-Medicaid expansion compared to pre-ARTS in VA compared to NC. Behavioral health acute inpatient admissions in VA decreased following ARTS implementation, and the decrease became larger after Medicaid expansion.

## 1. Introduction

Behavioral health disorders, including mental illnesses and substance use disorders (SUDs), continue to be a major public health concern in the United States [1]. An estimated 50% of Americans are diagnosed with a mental illness at some point in their lifetime [2]. In 2020, almost 23% of all adults were diagnosed with any mental illness (AMI) and 17% reported a SUD [3,4]. Statistics from the 2019 National Survey on Drug Use and Health indicate that 17 million adults have both a mental illness and a SUD [5].

To expand the scope of SUD treatment services for its members, Virginia (VA) Medicaid implemented the Addiction and Recovery Treatment Services (ARTS) benefit in April 2017 via a Centers for Medicare and Medicaid Services 1115 Demonstration Waiver [6]. ARTS expanded access to outpatient treatment services, Medications for Opioid Use Disorder (MOUD), short-term residential treatment, and inpatient detoxification services. ARTS also increased provider reimbursement rates for many existing services and introduced a new care delivery model, the Preferred Office-Based Opioid Treatment (OBOT) provider, which integrated MOUD with behavioral and physical health by incentivizing increased use of care coordination activities [7]. Furthermore, other behavioral health services were covered by existing managed care organizations, offering a more comprehensive care delivery system that further increases integration of addiction treatment services with other health services covered by Medicaid [1].

On 1 January 2019—nearly two years after the implementation of ARTS—VA expanded Medicaid eligibility for adults ages 19–64 with household incomes up to 138% of the federal poverty level. As of April 2023, about 754,000 low-income Virginians were enrolled through Medicaid expansion [8]. Medicaid expansion increases access to ARTS for many low-income adults who had a SUD prior to enrolling in Medicaid [9]. Virginia is unique in that a major expansion of the scope of Medicaid benefits for SUD treatment (the establishment of the ARTS program) in 2017 was followed by a major expansion in the breadth of eligibility (Medicaid expansion) in 2019. In other words, the increased scope of SUD treatment that was initially provided only to pre-expansion Medicaid beneficiaries in 2017 was extended to a larger population of low-income adults in 2019.

Previous studies showed mixed results on the effect of Medicaid expansion on the utilization of healthcare services in behavioral health. Some studies have reported increased access to inpatient behavioral services after Medicaid expansion [10,11]. However, Wen et al. found a reduction in opioid-related hospital admissions after Medicaid expansion [12]. Similarly, the likelihood of having an inpatient hospitalization declined among Medicaid beneficiaries with substance use disorders after ARTS implementation [13]. No study has examined the combination of expanded scope of benefits for behavioral health services followed by an expansion of breadth for eligibility expansion. Therefore, the main objective of this study is to evaluate the effects of expanding the scope and breadth of SUD treatment in Medicaid on the number of behavioral health acute inpatient admissions using a quasi-experimental design.

## 2. Materials and Methods

We examined trends in the number of acute inpatient admissions related to behavioral health disorders before and after the implementation of ARTS and of the Medicaid expansion in VA, with the study period between 2016 and 2019. We further use a difference-in-difference approach to examine the change in acute inpatient admissions for behavioral health relative to a comparison state, North Carolina (NC). Up to 2018, NC had a similar prevalence of mental health disorders and access to care measures to VA, including the prevalence of adults with AMI, any SUD in the past year, any serious thoughts of suicide, any AMI patients who are also uninsured, any AMI patients who did not receive treatment, any AMI patients reporting an unmet need, and any AMI patients who could not see a doctor due to costs [14]. These similarities in the prevalence of mental health and access to care measures strengthen the validity of using NC as a comparison state to evaluate policy effects in VA (Appendix A).

During the study period, NC did not expand Medicaid and did not implement changes in SUD treatment benefits for Medicaid similar to VA. Although NC implemented a Section 1115 SUD demonstration waiver in 2019, this waiver was limited to expanding payment for IMD (residential treatment services) and to transitioning from fee-for-service (FFS) to managed care but included no new additional benefits. Also, the changes included in the waiver were phased in over time and did not take full effect until after the study period [15].

### 2.1. Data Source

Our analysis uses inpatient administrative data from VA and NC. Virginia inpatient admissions data are obtained from Virginia Health Information’s (VHI) Patient-Level Data, while NC inpatient admissions are obtained from the Healthcare Cost and Utilization Project (HCUP) Central Distributor [16,17] (inpatient admissions from VA are not available through the HCUP Central Distributor, but only through VHI). Although there are some differences in definitions for some fields, the data sources are comparable in that they include all inpatient admissions for all acute-care hospitals and from all payers (including self-paid or uninsured) for each quarter of the year over the years 2016 to 2019. Both data sources include similar availability of information on the *International Classification of Diseases*, Tenth *Revision*, Clinical Modification (ICD-10-CM) codes for each behavioral health inpatient admission. The data sources are based on data from community hospitals, defined as short-term, non-federal, general, and other hospitals, excluding hospital units of other institutions (e.g., prisons), long-term care, rehabilitation, psychiatric, and alcoholism and chemical dependency hospitals. The analysis for this study is restricted to admissions to general, acute, short-term hospitals for problems related to behavioral health disorders [18].

### 2.2. Inclusion Criteria

We included inpatient admissions for all adults between 18 and 64 years admitted to the hospital with the diagnosis of mental illnesses and SUDs. Although the ARTS benefit focused on expanded benefits for SUD treatment, the high co-occurrence of SUD with mental illness [19] may affect behavioral health admissions more generally, especially since one of the intended benefits of ARTS was greater coordination with mental and physical health services [1]. Admissions are restricted to ages 18–64 years old because these are the age groups eligible for Medicaid expansion. To assess the impact of ARTS and Medicaid expansion on the population, we included individuals with all types of insurance coverage, including those who were uninsured (that is, whose admissions were self-paid or paid through hospital charity care programs) [12]. Including admissions for all payers and uninsured patients is especially important for assessing the effects of Medicaid expansion, since Medicaid admissions alone are likely to increase due to the increase in Medicaid eligibility, but overall admissions may decrease due to increased access to services for people who were previously uninsured. While it would have been optimal to further target the sample to admissions for people who were lower-income, information on patient income was not available in the admissions data.

### 2.3. Exclusion Criteria

We excluded psychiatric hospitals since they provide longer-term specialized care, while acute-care hospitals usually provide treatment to mentally ill patients for less than 30 days [20]. We focused on acute-care hospitals because the impacts of ARTS and Medicaid expansion on acute-care use were likely to be more immediate, whereas the impact on longer-term specialized care hospitals may take longer to detect [21]. In addition, the HCUP data source that was used for NC did not include psychiatric hospitals. So, this data limitation prevented us from including psychiatric hospitals in our analysis.

### 2.4. Description of Measures

The original data were admission-level; however, we aggregated the number of behavior health inpatient admissions for each county (and independent cities in VA) for county-level analysis based on county Federal Information Processing System (FIPS) codes [22]. Aggregation by county is based on the patient’s zip code of residence. Admissions for each county were aggregated by quarter, so that the unit of analysis is the county/quarter. For VA, there are 2128 observations—133 (95 counties and 38 independent cities) in 16 quarters (January 2016 through December 2019). For NC, there are 1600 observations—100 counties by 16 quarters.

### 2.5. Independent Variable

The main independent variables are time periods corresponding to the implementation of the ARTS program in VA in the second quarter of 2017 and the Medicaid expansion in the first quarter of 2019. A binary variable indicating whether the observation was in VA (coded as “1”) compared to NC is also included.

### 2.6. Dependent Variable

The primary dependent variable is the number of behavioral health acute inpatient admissions for each county by quarter defined based on ICD-10-CM codes (included in Appendix A) [23]. Behavioral health diagnosis could occur in any position in the claim, from primary to secondary diagnosis and beyond. All behavioral health disorders include any mental illness and SUD diagnoses. Admissions for mental illness include all admissions with a diagnosis of mental illness (with or without a secondary SUD diagnosis). Admissions for SUD include all admissions with a SUD diagnosis (with or without a secondary mental illness diagnosis).

### 2.7. Analytical Method

We performed descriptive analyses of the average number of behavioral health acute inpatient admissions by county (and independent city) from the first quarter of 2016 until the last quarter of 2019 in VA and NC.

We used quasi-experimental methods using Poisson fixed-effect event study regression to examine the number of behavioral health acute inpatient admissions in the quarters before and after ARTS and Medicaid expansion in VA and comparing these trends to the same quarters in NC. This analysis expands the difference-in-difference analyses by creating a separate parameter for each quarter of interest.

To control time-invariant characteristics of counties and independent cities, we included county-level fixed effects in all multivariate analyses. We also included time dummies and a quarter-specific measure of the percentage of uninsured patients under 65 years old as a time varying measure that might have an impact on our dependent variable in each quarter [12]. The measure of the uninsured percentage for people under 65 years old was obtained from the United States Census Bureau’s Small Area Health Insurance Estimates (SAHIE) Program [24].

The analytical model is specified as follows:Inpatient admission_it_ = β_0_ + Ψ_t_ + β_1_ (state_t_ × Ψ_t_) + x_it_ + α_i_ + u_it_

Ψ_t_ represents a full set of quarterly dummy variables (1 quarter is a reference + 15 dummy variables). State_t_ × Ψ_t_ represents the set of interactions between a dummy variable for VA and the quarterly dummies starting with the first quarter of 2016 to the fourth quarter of 2019 with indication of second quarter of 2017 as a reference when ARTS came into effect. x_it_ is the percentage of uninsured. α_i_ is the county fixed effect. u_it_ notates the error term.

By estimating the number of inpatient admissions through a Poisson model, we avoided the potential estimation bias that could result from estimating rates of admissions for behavioral health disorders [25]. For example, the total inpatient admissions might change (higher or lower) concurrent with the behavioral health admissions, making the behavioral health admissions’ rate biased. In addition, the Poisson model is appropriate for some counties where there is a low count of inpatient admissions [26].

We estimated the effect of the ARTS program on the number of behavioral health acute inpatient admissions after its implementation in April 2017 (11 quarters). Parameters for the interactions between the VA dummy variable and each quarterly time dummy variable estimated the proportional difference in admissions for VA relative to NC. We then averaged these estimates over different periods to assess the impact of ARTS and then the impact of ARTS combined with Medicaid expansion compared to the pre-ARTS period. We obtained linear combinations of parameters using the command “lincom” in Stata 13 [27] to compute the average estimate in the seven quarters after ARTS but before Medicaid expansion and then the last four quarters after Medicaid expansion. Thus, we estimate the average proportional difference in acute inpatient admissions in VA in these two periods relative to the time period preceding ARTS. We computed standard errors that are robust to heteroskedasticity for all multivariate models. The parameters of interest are the differences in the percentage change in VA acute inpatient admissions for behavioral health disorders relative to their level of acute inpatient admissions prior to implementation of ARTS minus the same difference for these type of acute inpatient admissions in NC. All models were estimated using Stata 13, and *p* < 0.05 was considered statistically significant. This study was exempted from receiving Institutional Review Board approval because the data were anonymized for public use.

## 3. Results

There was a total of 3728 county/quarter observations from 2016 to 2019. The average age of patients who were hospitalized for behavioral health disorders was 44 years for both VA and NC. The mean county average length of hospital stay for behavioral health disorders was four days in VA and five days in NC (Table 1).

The total acute inpatient admissions for behavioral health disorders were 713,513 in VA. Among these, 624,832 (87.57%) were for mental illnesses and 254,514 (35.67%) were for SUD (admissions that included both mental illness and SUD are included in both measures). In NC, there were a total of 1,092,067 acute inpatient admissions for behavioral health disorders, including 955,590 (87.50%) for mental illnesses and 385,166 (35.27%) for SUD (Table 1).

The unadjusted number of acute inpatient admissions per county per quarter for behavioral health disorders before and after ARTS implementation and after Medicaid expansion in VA and NC showed an increase in NC but virtually no change in VA (Table 2).

Results of the Poisson fixed-effect event study regression for the three dependent variables are shown in Table 3: (a) all behavioral health acute inpatient admissions, (b) acute inpatient admissions for mental illness only, (c) acute inpatient admissions for SUD only. The results show that following ARTS implementation, the number of behavioral health acute inpatient admissions decreased in VA relative to NC. The number of behavioral health acute inpatient admissions in the first quarter of 2018 decreased by 2.6% (95% CI [−5.1, −0.2]) compared to the second quarter of 2017 (the beginning of the ARTS program) in VA relative to NC. The reduction in behavioral health acute inpatient admissions remained statistically significant at 2.8% (95% CI [−5.3, −0.2]) in the second quarter of 2018, 4.2% (95% CI [−5.3, −0.2]) in the third quarter of 2018, and 4.9% (95% CI [−7.5, −2.4]) in the fourth quarter of 2018 compared to beginning of the ARTS program in VA relative to NC.

The slope of the change in behavioral health acute inpatient admissions was interrupted in the first quarter of 2019 when Medicaid expansion was implemented in VA (Figure 1). Following first quarter of 2019, decreases in admissions in VA relative to NC became larger, although only the decrease in the fourth quarter of 2019 was statistically significant at the 0.10 level (−3.8, CI [−7.6, 0.0]). Trends for mental illness admissions were similar following Medicaid expansion, while differences in SUD admissions were much smaller and not statistically significant (see Appendix A).

### Result of the Average Proportional Change in the Number of Acute Inpatient Admissions

To estimate the impact of ARTS alone and in combination with Medicaid expansion, we calculate the average of the parameters for the interaction of the VA dummy variable and the quarterly time dummies in the respective time periods. These averages reflect the proportional difference in acute inpatient admissions in VA compared to NC relative to the time period preceding ARTS. These estimates are presented in Table 4. In the seven quarters after ARTS but before Medicaid expansion, behavioral health acute inpatient admissions showed a significant decrease of approximately 2.7% (*p*-value 0.026). For mental illness acute inpatient admissions only, there was a 2.9% decrease (*p*-value 0.022). The decrease in SUD only acute inpatient admissions (−1.2%) was not statistically significant.

Table 4 also shows similar estimates for the period after Medicaid expansion (compared to the pre-ARTS period). The estimates of the average proportional difference are negative for mental illness acute inpatient admissions but positive for SUD acute inpatient admissions; however, these estimates are not statistically significant, indicating that the decrease in admissions in VA did not continue after Medicaid expansion. The last row of Table 4 also provides the difference of the average in the estimates between the two periods, which can also be interpreted as the incremental change in acute inpatient admissions following Medicaid expansion. The results show no statistically significant differences in the proportional change in acute inpatient admissions following Medicaid expansion compared to the time period after ARTS but before Medicaid expansion.

## 4. Discussion

In this study, we examined the effects of expanding the scope and breadth of SUD treatment in Medicaid on the number of acute inpatient admissions for behavioral health in VA compared to NC, which did not implement similar reforms during this time period. We compared acute inpatient admissions for behavioral health at the county level for each quarter for 2016 through 2019, assessing changes in acute inpatient admissions in VA after the implementation of the ARTS benefit in April 2017 and Medicaid expansion in January 2019, relative to changes in admissions in NC. We observed decreases in behavioral health acute admissions in VA in each of the seven quarters following ARTS implementation relative to NC. These decreases grew larger each quarter until the beginning of Medicaid expansion in VA in 2019.

The reduction in the number of acute inpatient admissions for behavioral health following the implementation of the VA ARTS program suggests that increasing the scope of outpatient addiction and residential SUD treatment services covered by Medicaid, as well as greater coordination and integration of SUD treatment with other mental health services, may have decreased the need for more intensive inpatient services. Since all the provisions of the ARTS benefit were implemented statewide on 1 April 2017, it is not possible to identify specific provisions of the ARTS benefit that were associated with decreases in acute inpatient admissions.

The reasons for the interruption of the trend in declining acute inpatient admissions related to behavioral health following Medicaid expansion are unclear. Acute inpatient admissions increased initially in 2019 for both VA and NC. The increase in behavioral health acute inpatient admissions in VA after Medicaid expansion may reflect in part pent-up demand for services among newly enrolled members who were previously uninsured and had undiagnosed SUD or mental illness, and therefore may have required higher level acute inpatient care when they first enrolled in Medicaid [28]. This potential explanation is consistent with prior research showing that expanding health insurance coverage increases utilization of all types, including inpatient services [11,29]. On the other hand, the fact that admissions increased in both VA and NC suggests the increase was not entirely driven by Medicaid expansion in VA. It is possible that the increase in admissions in 2019 reflects increases in SUD prevalence during this period in both VA and NC, as suggested by a surge in fatal overdoses beginning in 2019 in both states and continuing through 2020 and 2021 [30]. Ultimately, VA did not observe a statistically significant decrease in behavioral health acute inpatient admissions in the first year of Medicaid expansion relative to the pre-ARTS period, although the quarterly estimates for 2019 suggest such differences began to emerge in the fourth quarter of 2019.

In sum, the ARTS effect is likely due in part to increasing the scope of SUD treatment services for Medicaid members, while the Medicaid expansion effect may reflect in part initial pent-up demand for acute inpatient behavioral health care among those who were previously uninsured. It is possible that there is a greater lag in the effect of Medicaid expansion on admissions, and that such an effect would normally emerge in 2020 and 2021. However, the beginning of the COVID-19 pandemic in 2020 may have confounded these trends.

The findings in this paper are subject to some limitations. First, admissions data do not have information on many patient characteristics that might be most affected by Medicaid expansion, e.g., household income, educational attainment, housing stability, etc. We used the percentage of uninsured by county level as a proxy variable to address this issue. Second, ARTS and Medicaid expansion policies were not randomly assigned. Although our difference-in-differences methods control for some sources of bias, we cannot rule out the possibility that some other changes occurred differentially in VA compared to NC at the same time as these policies were adopted, potentially biasing our difference-in-differences estimates. However, we estimated event study regressions comparing changes between VA and NC to assess parallel trends in the pre-policy period, which showed that VA and NC had similar trends prior to the ARTS policy. Third, we used a single state to compare the effect of ARTS and Medicaid expansion in VA on behavioral health acute inpatient admissions, and therefore it is possible that the results would differ if other states were included as comparisons. However, NC is a reasonable comparison state with VA in terms of both geographical proximity and population characteristics. Finally, we analyzed the effect of Medicaid expansion in one year. However, studies showed the effect of Medicaid expansion on patients’ health outcomes was revealed after two years of implementation [21,31]. While the effects of Medicaid expansion may have continued and become more apparent in 2020 and 2021, the emergence of the COVID-19 pandemic in 2020 would make it difficult to observe the lagged effects of Medicaid expansion.

## 5. Conclusions

Behavioral health acute inpatient admissions steadily decreased in VA compared to NC following the implementation of the ARTS benefit. However, this trend was interrupted after the implementation of Medicaid expansion in VA in 2019. In VA, it is possible that initial pent-up demand among uninsured patients with undiagnosed SUD could have contributed to the increase in behavioral health acute admissions, while both states saw increases in fatal overdoses [32]. As with ARTS, there is some evidence of a lag in the Medicaid expansion “effect”, although it is difficult to follow this into 2020 due to the beginning of the COVID-19 pandemic. Policymakers should note that eligibility expansion may lead to different effects on acute inpatient admissions compared to the expansion of the scope of benefits covered for members.

## Figures and Tables

**Figure 1 ijerph-21-00777-f001:**
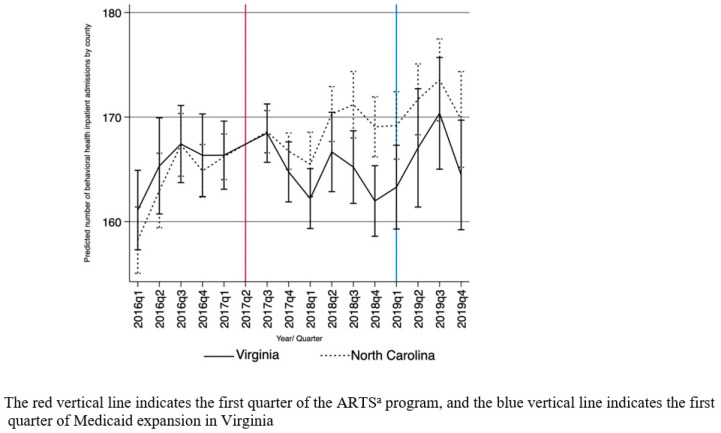
Impact of adoption of ARTS ^a^ and Medicaid expansion. Predicted number of behavioral health acute inpatient admissions by county/quarter. ^a^ Addiction and Recovery Treatment Services.

**Table 1 ijerph-21-00777-t001:** Mean characteristics of patients who were acutely admitted for behavioral health disorders.

	Virginia	North Carolina	*p*-Value
Total number of acute inpatient admissions for mental illness and SUD	713,513	1,092,067	<0.000
Total number of acute inpatient admissions for mental illness (%)	624,832 (87.6)	955,590 (87.5)	<0.000
Total number of acute inpatient admissions for SUD (%)	254,514 (35.7)	385,166 (32.3)	<0.000
Average age of admissions (years) (SD)	44.38 (1.7)	44.40 (1.3)	0.554
Average length of hospital stays (days) (SD)	4.30 (0.5)	4.67 (0.4)	<0.000
Average percentage of uninsured patients younger than 65 years old (SD)	10.24 (2.2)	13.10 (2.2)	<0.000

**Table 2 ijerph-21-00777-t002:** Mean number of quarterly county acute inpatient admissions before ARTS ^a^, between ARTS and Medicaid expansion, and after Medicaid expansion.

Virginia	Pre-ARTS (5 Quarters)	Between ARTS and Medicaid Expansion (7 Quarters)	Post-Medicaid Expansion (4 Quarters)
Total admissions (N)	797	788	780
All behavioral health acute inpatient admissions (%)	334 (41.9)	335 (42.5)	337 (43.2)
Mental illness acute inpatient admissions (%)	293 (36.8)	293 (37.2)	295 (37.8)
SUD acute inpatient admissions (%)	117 (14.7)	119 (15.1)	123 (15.8)
North Carolina			
Total admissions	1326	1320	1318
All behavioral health acute inpatient admissions (%)	666 (49.1)	685 (51.9)	700 (53.1)
Mental illness acute inpatient admissions (%)	582 (43.9)	599 (45.4)	613 (46.5)
SUD acute inpatient admissions (%)	234 (17.7)	241 (18.3)	249 (18.9)

^a^ Addiction and Recovery Treatment Services.

**Table 3 ijerph-21-00777-t003:** The poison regression percentage change (%) in behavioral health acute inpatient admissions in Virginia compared to North Carolina after and before ARTS ^a^.

VA/Year–Quarter	All Behavioral Health Acute Inpatient Admissions 95% (CI)	Mental Illness Acute Inpatient Admissions 95% (CI)	SUD Acute Inpatient Admissions 95% (CI)
2016Q1	1.2 (−1.7, 4.0)	1.2 (−1.8, 4.2)	0.7 (−3.5, 4.9)
2016Q2	0.7 (−2.6, 4.1)	0.5 (−3.1, 4.1)	−1.0 (−5.8, 3.8)
2016Q3	−0.6 (−3.4, 2.2)	−1.1 (−4.1, 1.8)	0.6 (−3.3, 4.5)
2016Q4	0.3 (−2.4, 2.9)	1.0 (−2.6, 2.8)	−0.4 (−4.5, 3.8)
2017Q1	−0.6 (−2.7, 1.7)	−0.5 (−2.8, 1.8)	−1.7 (−5.4, 2.0)
2017Q2	Reference	Reference	Reference
2017Q3	−0.7 (−2.6, 1.1)	−1.7 (−3.7, 0.3)	1.5 (−1.5, 4.5)
2017Q4	−1.8 (−3.7, 0)	−2.3 * (−4.4, −0.2)	−0.8 (−4.0, 2.4)
2018 Q1	−2.6 * (−5.1, −0.2)	−3.0 * (−5.4, −0.4)	−1.8 (−6.2, 2.5)
2018Q2	−2.8 * (−5.3, −0.2)	−3.5 * (−6.3, −0.8)	−3.2 (−7.2, 0.8)
2018Q3	−4.2 ** (−5.3, −0.2)	−4.7 ** (−7.5, −1.9)	−2.9 (−7.4, 1.6)
2018Q4	−4.9 *** (−7.5, −2.4)	−5.3 *** (−8.2, −2.4)	−4.0 (−8.5, 0.6)
2019Q1	−1.7 (−5.1, 1.7)	−1.9 (−5.4, 1.6)	1.3 (−4.5, 7.2)
2019Q2	−3.4 (−6.9, 0.2)	−4.0 (−7.7, −0.3)	0.5 (−5.6, 6.5)
2019Q3	−2.5 (−6.0, 1.0)	−2.8 (−6.4, 0.9)	1.3 (−4.5, 7.1)
2019Q4	−3.8 * (−7.6, −0.0)	−4.6 * (−8.5, −0.7)	−0.2 (−5.8, 5.3)

^a^ Addiction and Recovery Treatment Services. * *p* < 0.05, ** *p* < 0.01, *** *p* < 0.001.

**Table 4 ijerph-21-00777-t004:** Average of quarterly parameter estimates in different periods estimating the proportional change in the number of acute inpatient admissions in Virginia relative to North Carolina.

	Behavioral Health Acute Inpatient Admissions	*p*-Value	Mental Illness Acute Inpatient Admissions	*p*-Value	Substance Use Disorder Acute Inpatient Admissions	*p*-Value
After ARTS, before Medicaid expansion (7 quarters), 95% (CI)	−2.5 (−4.1, −0.8)	0.003	−2.9 (−4.7, −1.2)	0.001	−1.6 (−4.4, 1.2)	0.256
After Medicaid expansion (4 quarters), 95% (CI)	−2.9 (−6.1, 0.4)	0.089	−3.33 (−6.8, 0.1)	0.057	0.7 (−4.6, 6.1)	0.792
Difference, 95% (CI)	0.4 (−1.9, 2.7)	0.735	0.41 (−2.0, 2.8)	0.735	−2.3 (−5.8, 1.2)	0.195

## Data Availability

The original data presented in the study are openly available in Patient-Level Data Virginia, https://www.vhi.org/Products/patientleveldata.asp (accessed on 23 October 2023), and in the Healthcare Cost and Utilization Project (HCUP) https://www.ahrq.gov/data/hcup/index.html (accessed on 23 October 2023).

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
