# Peer review of "The Cumulative Effect of Expanding the Breadth and Scope of Coverage for Substance Use Disorder Treatment on Behavioral Health Acute Inpatient Admissions: Evidence from Virginia Medicaid"

_ijerph, 2024, doi:10.3390/ijerph21060777_

Round 1

Reviewer 1 Report

Comments and Suggestions for Authors

Author Response

Dear Colleague,

We appreciate the review and thoughtful suggestions and comments. We worked hard to address all of the comments. We hope that you will find our revisions acceptable.

Reviewer 1:

The authors take up an original and well-defined approach to an important policy question—the impact of expanding and improving SUD services and expanding who has access to publicly financed SUD services broadly and Virginia ARTS and Medicaid expansion in Virginia specifically. The results advance current knowledge and would be beneficial to many people interested in these questions both in the United States and other countries with publicly financed SUD treatment. The work seems to fit the journal scope but I am not a regular reader of the International Journal of Environmental Research and Public Health so I’m not in an ideal position to judge this question. The quality of the presentation is high. The use of English language is good and the article is written in an appropriate way. The data and analyses are presented appropriately except where I indicate below that it would be helpful to add a table note.

I review by manuscript section:

Abstract

  •  I suggest the authors provide some detail on what “expanding the scope and breadth” entailed.

Author response: Thank you for your comment. We revised the abstract:

 We evaluated the impact of two Medicaid policies in Virginia (VA), the Addiction and Recovery Treatment Services (ARTS) program and Medicaid expansion on the number of behavioral health inpatient admissions from 2016 to 2019. We used Poisson fixed-effect event study regression and compared average proportional differences in admissions over three time periods: (1) prior to ARTS; (2) following ARTS but before Medicaid expansion; (3) post-Medicaid expansion. The number of behavioral health inpatient admissions decreased by 2.6% (95% CI [-5.1, -0.2]) in the first quarter of 2018 and this decrease gradually intensified by 4.9% (95% CI [-7.5, -2.4]) in the fourth quarter of 2018 compared to the second quarter of 2017 ( beginning of ARTS) in VA relative to North Carolina (NC). Following the first quarter of 2019 ( beginning of Medicaid expansion), decreases in VA admissions became larger relative to NC. The average proportional difference estimated a decrease of 2.7% (95% CI, [−4.1,−0.8])  after ARTS but before Medicaid expansion and a decrease of 2.9% (95% CI, [−6.1, 0.4]) post-Medicaid expansion compared to pre-ARTS in VA behavioral health inpatient admissions compared to NC. Behavioral health inpatient admissions in VA decreased following ARTS implementation, and the decrease became larger after Medicaid expansion.

  •  Spell out “VA” and “NC”.

Author response: Thank you for your comment. We spelled out both VA and NC in the abstract.

Introduction

  • I don’t think international readers will know what “carve in” to MCOs means. I suggest you briefly elaborate.

Author response: Thank you for your comment. We changed the “carved in” to “Covered by” to make it more clear for international readers.

Materials and Methods

  •  I suggest the authors add a note for Table 1. It’s unclear where the data came from

Author response: Thank you for your comment. We replaced Table 1 in the result section since it belongs to our findings and used the same dataset as the rest of the analysis.

  •  Data Source—

I haven’t worked with these data but based on similar data I use, they seem suitable for the analysis and robust enough for the authors’ conclusions.

Author response: Thank you for your feedback.

  •  Inclusion Criteria—I’m confused about why the authors “included inpatient admissions for all adults between 18 and 64 years admitted to the hospital with the diagnosis of mental illnesses and SUD”. Based on the rest of the text I think it should be SUD or MI? If it’s not a typo, the authors should expand on why they chose this study population.

Author response: Thank you for your comment. We replaced “ And” to “or” in the sentence “included inpatient admissions for all adults between 18 and 64 years admitted to the hospital with the diagnosis of mental illnesses or and SUD.”  We chose adults with SUD and Mental illnesses because ARTS focused on the expanded benefits of SUD treatment. However, the high co-occurrence of SUD with mental illness may affect behavioral health admissions more generally, especially since one of the intended benefits of ARTS was greater coordination with mental and physical health services. Admissions are restricted to ages 18–64 years old, because they are the age groups eligible for Medicaid expansion.

  •  Exclusion Criteria—I’m also confused why psychiatric hospitals were excluded from all analyses. I understand that you want to focus on acute care but ARTS could also impact longer term psychiatric care. It would be good to more clearly indicate that the authors focused on the impact on acute care (e.g., add “acute-care” before IP in the Abstract and Material and Methods sections), expand on your rationale, and/or conduct a sensitivity analysis that includes psychiatric hospitals.

Author response: Thank you for your comment. We focused on acute care hospitals because the impacts from ARTS and Medicaid expansion on acute care use were likely to be more immediate, whereas the impact on longer-term specialized care hospitals may take longer to detect. We included “acute” before” inpatient admission” throughout the manuscript. In addition, the HCUP data source that was used for NC didn't include psychiatric hospitals. So, the data limitation prevented us from including psychiatric hospitals in our analysis.  

  •  Description of Measures—This seems reasonable.

Author response: Thank you for your feedback.

  •  Independent Variables—I suggest the authors consider noting as a limitation that there’s a lag between the IVs as defined and as they theoretically operate—i.e., it takes some time for patients to respond to receipt of ARTS SUD care and many expansion enrollees do not enroll when expansion occurs or find care when they do enroll. This will weaken effects as the authors allude to in the Discussion.

Author response: Thank you for your comment. We expanded this limitation in the discussion:

Finally, we analyzed the effect of Medicaid expansion in one year. However, studies showed the effect of Medicaid expansion on patients’ health outcomes was revealed after two years of implementation.  While the effects of Medicaid expansion in our study may have continued and become more apparent in 2020 and 2021, the emergence of the COVID-19 pandemic in 2020 would make it difficult to observe the lagged effects of Medicaid expansion.”

  •  Dependent Variables—The SUD and MI definitions mostly look reasonable but it’s unclear why the authors included the codes under “Screening and history of mental health and substance abuse codes”. I suggest the authors explain why these codes are a treatment need at admission or run the analysis without them and re-do the exhibits if it has a more than negligible impact.

Author response: Thank you for your comment. First-time patients with symptoms of SUD or mental disorders such as major depression should be screened by diagnostic tests or tools. After the diagnosis, the patient might be admitted to the hospital based on the physicians’ decision. The final diagnosis is coded as secondary or beyond diagnosis. In this study, all of the patients with the “Screening and history of mental health and substance abuse codes” had SUB or mental illnesses that required inpatient admission.  Some other patients might need to be hospitalized for their condition; however, they self-reported the history of their condition without extra tests or screening tools.

Analytical Method

  •  The study is generally well designed and technically sound. However, it’s unclear how data from counties and independent cities relate and what the analysis did with them. I suggest the authors briefly elaborate. I am assuming that cities are nested within counties so it’s confusing why they’re in the analysis.

Author response: Thank you for your comment. Virginia is divided into 95 counties, along with 38 independent cities that are considered county-equivalents  for census purposes, totaling 133 second-level subdivisions. In Virginia, cities are co-equal levels of government to counties, but towns are part of counties. 

  •  The statistical methods seem reasonable but statistical methods are not my expertise.

  Author response: Thank you for your feedback.

Results

  •  The authors take a reasonable interpretation of the results.

Author response: Thank you for your feedback.

Discussion

  •  The authors appropriately discuss the results and offer cautious interpretation.

Author response: Thank you for your feedback.

  •  I suggest the authors touch on what is known in the literature about lags in effects from expanded services and expanded enrollment.

Author response: Thank you for your comment. We include two citations in the lag of Medicaid expansion effect on patients’ health outcomes: References [21] and [31]

  •  As noted above, the authors should consider adding the timing of the IVs as a limitation since they do not have data on patterns of ARTS usage and expansion enrollee use of behavioral health care.

Author response: Thank you for your comment. We expanded the limitation as “Finally, we analyzed the effect of Medicaid expansion in one year. However, studies showed the effect of Medicaid expansion on patients’ health outcomes was revealed after two years of implementation.  While the effects of Medicaid expansion in our study may have continued and become more apparent in 2020 and 2021, the emergence of the COVID-19 pandemic in 2020 would make it difficult to observe the lagged effects of Medicaid expansion.”

Conclusions

  • Conclusions are justified and supported by the results.

                  Author response: Thank you for your feedback.

Reviewer 2 Report

Comments and Suggestions for Authors

I read with interest the paper titled "The Cumulative Effect of Expanding the Breadth and Scope of Coverage for Substance Use Disorder Treatment on Inpatient Admissions: Evidence from Virginia Medicaid"

The paper is well written, however I have some minor comments to do before acceptance:

1) Abstract should focus more on the results of the paper. It seems it focus too much on explaining the methods, and less in sharing what you found in the manuscript. 

2) Data intervals should be well explained. In abstract you have 2016-2019 data; but then in the methods is 2016-2020. Please clarify if there are the same or different things. 

3) Table 1 is a result, since it have statistical tests. It seems doesn't need to be within the methods section. 

4) Please clarify the signals in table 3. A lot of signals "-" are used to express negative values, but also to present the distance between the interval, which make it confuses to read. You can wither add a column for upper and lower limit, or write differently.

Author Response

Dear Colleague,

We appreciate the review and thoughtful suggestions and comments. We worked hard to address all of the comments. We hope that you will find our revisions acceptable.

Reviewer 2:

Comments and Suggestions for Authors

I read with interest the paper titled "The Cumulative Effect of Expanding the Breadth and Scope of Coverage for Substance Use Disorder Treatment on Inpatient Admissions: Evidence from Virginia Medicaid"

The paper is well written, however I have some minor comments to do before acceptance:

  • Abstract should focus more on the results of the paper. It seems it focus too much on explaining the methods, and less in sharing what you found in the manuscript. 

Author response: Thank you for your comment. We revised the abstract to include more study findings:

We evaluated the impact of two Medicaid policies in Virginia (VA), the Addiction and Recovery Treatment Services (ARTS) program and Medicaid expansion on the number of behavioral health inpatient admissions from 2016 to 2019. We used Poisson fixed-effect event study regression and compared average proportional differences in admissions over three time periods: (1) prior to ARTS; (2) following ARTS but before Medicaid expansion; (3) post-Medicaid expansion. The number of behavioral health inpatient admissions decreased by 2.6% (95% CI [-5.1, -0.2]) in the first quarter of 2018 and this decrease gradually intensified by 4.9% (95% CI [-7.5, -2.4]) in the fourth quarter of 2018 compared to the second quarter of 2017 ( beginning of ARTS) in VA relative to North Carolina (NC). Following the first quarter of 2019 ( beginning of Medicaid expansion), decreases in VA admissions became larger relative to NC. The average proportional difference estimated a decrease of 2.7% (95% CI, [−4.1,−0.8])  after ARTS but before Medicaid expansion and a decrease of 2.9% (95% CI, [−6.1, 0.4]) post-Medicaid expansion compared to pre-ARTS in VA behavioral health inpatient admissions compared to NC. Behavioral health inpatient admissions in VA decreased following ARTS implementation, and the decrease became larger after Medicaid expansion.

  • Data intervals should be well explained. In abstract you have 2016-2019 data; but then in the methods is 2016-2020. Please clarify if there are the same or different things. 

Author response: Thank you for your feedback. The analysis was between 2016 and 2019, so the time period in the method section has been changed to 2016-2019.

  • Table 1 is a result, since it have statistical tests. It seems doesn't need to be within the methods section. 

Author response: Thank you for your comment. Table 1 belongs in the result section, so we moved it there.

  • Please clarify the signals in table 3. A lot of signals "-" are used to express negative values, but also to present the distance between the interval, which make it confuses to read. You can wither add a column for upper and lower limit, or write differently.

Author response: Thank you for your comment. We replaced “–“ with, “,” to show the distance between the upper and lower levels of CI.
